# Aging and the (Chemical) Senses: Implications for Food Behaviour Amongst Elderly Consumers

**DOI:** 10.3390/foods10010168

**Published:** 2021-01-15

**Authors:** Charles Spence, Jozef Youssef

**Affiliations:** 1Crossmodal Research Laboratory, Department of Experimental Psychology, University of Oxford, Anna Watts Building, Oxford OX2 6GG, UK; 2Kitchen Theory, Unit 9A Alston Works, London EN5 4EL, UK; jozef@kitchen-theory.com

**Keywords:** aging, chemical senses, multisensory flavour perception, gustation, olfaction, digital commensality

## Abstract

The growing aging population are increasingly suffering from the negative health consequences of the age-related decline in their senses, especially their chemical senses. Unfortunately, however, unlike for the higher senses of vision and hearing, there is currently nothing that can be done to bring back the chemical senses once they are lost (or have started their inevitable decline). The evidence suggests that such chemosensory changes can result in a range of maladaptive food behaviours, including the addition of more salt and sugar to food and drink in order to experience the same taste intensity while, at the same time, reducing their overall consumption because food has lost its savour. Here, though, it is also important to stress the importance of the more social aspects of eating and drinking, given the evidence suggesting that a growing number of older individuals are consuming more of their meals alone than ever before. Various solutions have been put forward in order to try to enhance the food experience amongst the elderly, including everything from optimising the product-intrinsic food inputs provided to the remaining functional senses through to a variety of digital interventions. Ultimately, however, the aim has to be to encourage healthier patterns of food consumption amongst this rapidly-growing section of the population by optimising the sensory, nutritional, social, and emotional aspects of eating and drinking. An experimental dinner with the residents of one such home where nostalgic-flavoured healthy ice-creams were served is described.

## 1. Introduction

The chemical senses, namely smell (olfaction), taste (gustation), and the trigeminal sense, just like the higher spatial senses of vision, audition, and touch (e.g., [1,2,3,4]) start their inevitable decline as we age (e.g., [5,6,7,8,9,10,11,12,13,14,15]). Unfortunately, however, there are currently no remediation devices, such as glasses and hearing aids, that can be used to make up for the loss of the chemical senses [16,17]. This is especially problematic given a growing aging population [18], with more people than ever before living to an age where impairments to their chemical senses starts to become much more noticeable. To give some sense of the emerging problem, according to the National Institute on Aging, National Institutes of Health [19], the number of those living into their eighties has grown exponentially over the preceding 40 years.

### 1.1. Sensory Decline in Aging

Our sensory and perceptual functions start their inevitable decline at different ages and at very different rates (see [20,21,22]). For example, while tactile, auditory and visual perceptual abilities show clear signs of decline by the time we reach middle age, taste and smell sensitivity do not show any marked deterioration until we reach 60–70 years of age, whereupon aging takes a greater toll on smell than on taste [2,15,23,24,25,26]. These age-related declines in sensory processing reflect the consequences of both peripheral physical and central neural degeneration (such as the increasing opacity of the lens of the eye, and the general reduction in the population of nerve cells), as well as the cognitive decline associated with a general loss of flexibility of mental processing in older individuals (see also [27,28,29]). Some commentators have even suggested that this cognitive inflexibility may actually provide older people with the “comfort and security in seeing and hearing events in the accustomed way” ([2], p. 96; see also [30]).

The decline in the sense of smell with aging [26,31,32] is likely to exert a more pronounced detrimental effect on multisensory flavour perception than any loss of taste (gustation), given figures suggesting that as much as 75–95% of what we think we taste, we actually smell [33]. Here, though, it is important to highlight the potentially important distinction between orthonasal smell (as when we sniff food) and the retronasal release of volatile-rich aromas pulsed out from the back of the nose when we swallow and masticate [34,35]. Interestingly, while most studies of the decline in olfactory sensitivity with aging have assessed orthonasal olfaction, it has been suggested that the findings do not necessarily provide a reliable prediction of retronasal experience, especially when dealing with complex foods [9,36,37,38]. Research looking specifically at the age-related loss in the sense of taste (gustation) has revealed a decline in sensation for the majority of basic taste qualities [39,40,41]. That said, there is also some older evidence to suggest that people’s sensitivity to specific individual taste qualities may also change differentially across the lifespan. For example, the sensitivity to sweetness is sometimes maintained while the sensitivity to saltiness has been reported to deteriorate significantly (e.g., [42]).

One of the key problems for those hoping to optimise the design of food and drink products for the elderly is that the variability in sensitivity to olfactory stimuli across the population tends to increase as we age [43]. What this means, in practice, is that while some elderly individuals may be functionally anosmic, others may retain a level of olfactory functioning that is not much different from their younger counterparts. Another point to stress here is that, if anything, the decline in chemosensory function in the elderly often appears to be more apparent when assessed with pure tastants/olfactory stimuli than when assessed with real food stimuli (e.g., see [40,41]). Here, it would certainly also be interesting to know more about whether or not orthonasal and retronasal olfactory abilities decline at the same rate in the elderly [9,44].

Intriguingly, those who have lost the ability to taste, for example, in the case of herpes zoster oticus in the case of the psychophysicist Pfaffmann (see [45]), report remarkably little loss of taste sensation. Similarly, Brillat-Savarin [46] reported on the case of soldiers whose tongues had been cut out in the Algerian war also reporting little loss of sensation as far as the flavour of food and drink were concerned. By contrast, those of us who have had a head cold know only too well the profound loss of taste that often results when olfactory inputs are not available.

Meanwhile, in a study of nearly 2000 people ranging from 5 to 99 years in age, Doty et al. [24] reported that the ability to identify smells peaked between the ages of 20 and 40 years, and started to decline thereafter. In fact, over half of the 65–80 year olds tested by Doty and his colleagues, and more than three-quarters of those over 80 years of age, exhibited major impairments in olfactory processing, with many exhibiting a clinical deficit in their ability to sense, known as anosmia (see also the National Geographic Smell Survey, [47], for a much larger survey of the effects of age on smell; and [5]). Crucially, recent increases in life expectancy mean that more people than ever before are currently suffering from age-related impairments in their ability to taste and smell. To give some idea of the magnitude of this change in chemosensory perception, it is predicted that by the year 2025 more than a billion people around the world will be over 60 years of age [18,48].

Research directly comparing age-related declines in olfactory and gustatory sensitivity suggests that olfactory losses tend to start earlier and to be more severe than those seen for taste (e.g., [7]). Moreover, it is important to note that the severity and nature of these age-related declines in chemosensory functioning show marked variability across individuals, and also vary markedly as a function of the particular measures of olfactory perceptual ability that happen to be used (e.g., [14]). The decline in chemosensory functioning presumably helps to explain why it is that so many elderly individuals complain that food lacks flavour (given that much of the flavour of food actually comes from its smell; [8]). The growing popularity of pungent spices, such as chilli pepper, black pepper, and ginger in food has also been attributed in part to the effects of aging [49,50]. Similarly the provision of a variety of seasonings, such as butter, tomato ketchup, lemon, parsley, mayonnaise, etc., was shown to exert a positive effect on meal enjoyment and food intake in one study conducted in a nursing home [51,52].

### 1.2. Decline in Salivary Function with Aging

Although it is not often mentioned in the literature, it is important to note that saliva also plays a key role in helping us to masticate and swallow food (see [53], for a review). It also plays an important role in our ability to experience the taste and flavour of food too [54,55]. Importantly, salivary function has been reported to decline with increasing age and hence this will also interfere with multisensory flavour perception in the elderly (see [56,57,58,59,60,61]). The effect of aging on the gut should not be forgotten either [62].

## 2. Unhealthy Eating Habits in the Elderly

The declining chemosensory abilities that have been extensively documented in the elderly can all too easily lead to unhealthy eating habits [63,64], as the latter increase their intake of salt and sugar to make up for their inability to taste these ingredients in food at lower concentrations (e.g., [16,65]). According to Stevens, Cain, Demarque, and Ruthruff [66], older individuals may need to add as much as two or three times more salt to perceive the same intensity in a tomato soup as those who are younger. Shockingly, this figure increased to twelve times for those older individuals who were on five or more medications, which turns out to be the majority of them (see also [8]). Given the negative health consequences of the overconsumption of salt (e.g., hypertension), this likely represents a very serious issue, and one that needs to be tackled by those hoping to optimise food delivery amongst the elderly. Note here only that according to observational data from the Framingham study in North America, the lifetime risk of developing hypertension in those who are 55–65 years of age is 90% [67].

Researchers argue that many of the most serious problems faced by the elderly stem from age-related changes in their sense of smell and to a lesser extent taste, rather than from the age-related declines affecting any of the other senses. This is all the more unfortunate given that many older people claim that eating and drinking represent the last remaining pleasures in their lives [68]. Or, as the famous French gastronome Jean Anthelme Brillat-Savarin put it almost two centuries ago that: “The pleasures of the table, belong to all times and all ages, to every country and to every day; they go hand in hand with all our other pleasures, outlast them, and remain to console us for their loss.” ([46], p. 14).

The fundamental point here is that the provision of acceptable food needs to be recognised as an important factor determining the quality of life for many older individuals, especially those who find themselves living in care facilities ([69], p. 150; [70]).

### Malnutrition—An Increasingly Common Problem Amongst the Elderly

Many older people, especially those who find themselves in hospital or else in a care home setting, all too often fail to eat sufficiently to maintain their weight (e.g., [71,72,73,74]). It is therefore critically important that we introduce strategies to encourage increased consumption of nutritionally-balanced foods and so avoid the malnutrition that is so often reported amongst the elderly (see also [75]). Indeed, the importance of establishing robust solutions for long-term care residents has been highlighted by a number of authors in recent decades (e.g., [76]; see also [77]). Meanwhile, according to another recent report from the National Health Service here in the UK, providing elderly patients with an extra meal a day halved their chances of dying while in hospital [78]. This presumably assumes that they eat that meal, and do not return it to the kitchen untouched, as is unfortunately so often the case [73]. One-on-one support with eating at mealtimes has been shown to be especially effective in increasing patients’ consumption of food in the hospital setting, but the associated staff costs normally limit the uptake of this kind of solution [79]. The metallic taste that many patients report while undergoing treatment for cancer would also appear to be an effective appetite suppressant (see [80]). While cancer can strike at any age, the risk increases markedly with age [81,82], thus meaning that the problem of metallic taste may be more prevalent amongst this age group.

## 3. Enhancing the Sensory Appeal of Food and Drink Amongst the Elderly

One route to increasing the sensory/perceptual interest of food and drink amongst older populations is to enhance the stimulation provided by the remaining functional senses [83,84]. It has, for example, been suggested that increasing the trigeminal input by incorporating more pepper pungency, heat from chili, and ginger, etc., may help to prevent foods from becoming too bland as an individual’s gustatory and olfactory function declines ([49]; though see also [85]; and [86], on the effects of interactions between texture and trigeminal stimulus in a liquid food system on the preferences of elderly consumers). That said, the evidence to date demonstrating increased consumption of taste/flavour-enhanced foods amongst the elderly is mixed (see [87], for a review of research to date).

When thinking about how to enhance the design of food and drink experiences amongst the elderly, it is important to note that much of our multisensory flavour experience is determined by our flavour expectations that are built on the basis of associative learning as a result of our prior food experiences. The latter tend to be set by sight, orthonasal smell (sniffing), and, on occasion, sound (think here only of the sizzle of the steak on the hot plate, or the ding of the microwave; [88]) and touch cues—as we feel the softness of the fruit, or the heat emanating from the outer surface of our coffee cup (e.g., [89]). It has been suggested that very often, we live in the world of our flavour expectations, only occasionally checking on the taste of what we are actually consuming (see [90,91], for reviews). Should the taste experience be pretty much as we expected, then we largely live in the world of our flavour expectations. If, however, there is too much of a divergence between expectations and experience, then this may very well lead to a negatively-valenced disconfirmation of expectation response [92,93]. At the same time, however, it is also important to recognize how mental imagery sometimes help to fill in the gaps in our perception of what we expect to taste and smell (see [94,95]). The key point here though is that the visual attributes of those food products designed specifically for the elderly should not be neglected. That is, it is not enough to simply think about flavour enhancement as the only solution to get elderly individuals to eat more/better [87].

It has long been suggested that meals should be made more colourful and sonically interesting for elderly and hospitalized individuals (e.g., [79,96]). Further, beyond enhancing the multisensory perception of the food itself, one should not neglect the importance of the colour of the plateware on which it is served (see [97] for a review of the impact of plateware colour on taste and consumption). It has been reported that some older individuals may struggle to distinguish the food from the plate visually, especially when pallid white institutional foods are served against the background of the ubiquitous large round white American plate. Interesting here, therefore, is research showing that simply by switching to high-contrast coloured (e.g., red or blue) plateware and glassware, a dramatic increase in the consumption of food can be achieved, at least in the short term, amongst older patients and care home residents (e.g., [98,99,100,101,102]). Thinking more carefully about the presentation of the food can also help to encourage greater consumption amongst the elderly [103].

Many older individuals, including those who have lost their teeth, are often fed pureed meals as they can find it hard to deal with solid foods [104]. Unfortunately, however, such texturally monotonous foods have lost most of their sensorial interest and hence may result in undernutrition. What is more, the absence of textural cues make it much harder for people to identify vegetables too (see [105]). Note here also how many elderly people suffer from a more general difficulty in identifying foods [106]. This is potentially important because people are less likely to consume those foods that they struggle to identify.

Japanese researchers have developed a headset that older people in this situation can wear that presents mastication-like sounds elicited by jaw movements [107,108]. It has even been suggested that different sounds might be used to represent different food textures, thus providing an additional sensory cue. Preliminary findings suggest that this might provide an effective means of adding some sonic interest to mealtimes for such individuals, it remains to be seen whether there will be widespread uptake of such high-tech solutions, especially amongst elderly and hospitalized individuals.

## 4. Ice-Cream as an Effective Vehicle for Nutrient Delivery in the Elderly

Ice-cream is often noted as being a popular food amongst many older individuals (e.g., [109,110]). Indeed, its unique sensory properties have been highlighted as one of the reasons why so many people find that they still have space for this highly-desirable food even when they are otherwise full at the end of the meal ([111]; see also [112,113]). It may be the case that in some older individuals, the oral-somatosensory cues that are provided by the cold temperature of the ice-cream, as well as perhaps the fatty/creamy mouthfeel characteristics help to provide agreeable sensory stimulation from food in those who may otherwise be suffering from marked olfactory loss or else may even be functionally anosmic (e.g., [47]).

Working with this idea, and against the common conception of ice-cream as an unhealthy (and possibly childish/necessarily indulgent) food, chef Jozef Youssef of Kitchen Theory [114] created a series of ice-creams using a range of healthier ingredients (including pureed vegetables and meal replacement powders such as Huel; [110,115]). Furthermore, by using a Pacojet machine to make the ice-cream, it was possible to deliver a deliciously-smooth texture without the necessity of adding cream. Unfortunately, however, the high price of the latter machines (c. £5000 for a new model) will likely limit the uptake of this item of modernist culinary technology by most of those providing food for the elderly. The intervention of chefs and neuroscientists to help recover pleasure lost due to sensory loss was also investigated by the Roca Brothers in Spain in 2019 (see [116,117], for a couple of press reports).

While the concept of savoury ice-creams is currently unfamiliar to many Western consumers, they are nevertheless popular in Japan, as well as in the context of many modernist restaurants around the world [118,119,120]). What is more, savoury ice-creams were once popular in Europe and presumably beyond during the 18th and 19th centuries (e.g., see [121,122]). Appropriately-designed (i.e., nutritionally-balanced) ice-creams may therefore provide an excellent vehicle for the delivery of protein and other elements necessary for a balanced diet in elderly populations who might otherwise be suffering from poor nutrition [123,124,125,126]; see also [127,128]). Here, it is worth noting that much the same approach to nutritionally-enhanced ice-cream has also been proposed previously in the case of cancer patients [129]. That said, when introducing novel ice-cream flavours, one has to be careful not to trigger a negatively-valenced ‘disconfirmation of expectation’ response amongst consumers, elderly or otherwise, who may initially be unfamiliar with such savoury flavours in the context of ice-cream ([130]; see also [92,131]).

Chef Jozef Youssef and his team created a range of savoury ice-creams with various flavours chosen to elicit positive nostalgia amongst older individuals. Given the UK base for this particular intervention, the meal incorporated Heinz cream of tomato soup, prawn cocktail, and bone marrow ice cream flavours (see Figure 1 for the menu from the event). These dishes were served in the context of a multisensory environment that was itself designed to trigger positive nostalgia. For example, traditional visual designs were projected onto the dining table along with retro food labels matching the flavour of the dish the aged diners, and their carers, were currently eating using projection mapping (see Figure 2). Meanwhile, Vera Lynn, one of the most popular vocalists during the war years, and such-like was presented over the loudspeakers. Several recent studies have demonstrated that ambient soundscapes influence both the sensory-discriminative and hedonic experience of ice-cream and gelati [132,133,134,135,136]. Although no quantitative data were obtained, the qualitative reports of the various residents and supporters of the Denville Hall residential home for aged actors who supported this particular culinary exercise were, on the whole, very positive [137]. Indeed, the authors hope to collect quantitative data to support the approach outlined here (namely using healthy ice-cream as a vehicle to enhance food consumption behaviours amongst the elderly) in the near future. Until such time, however, the findings reported here should be treated as merely anecdotal.

### 4.1. Music and Soundscapes to Enhance Meal Times Amongst Agitated Seniors

Another relatively-simple low-cost intervention to enhance food behaviour at mealtimes is to use music, or ambient soundscapes, to help relax those individuals who might otherwise be too agitated to eat. This is apparently a common problem amongst many psychiatric patients as well as a growing number of those older individuals who are suffering from Alzheimers/dementia (e.g., [69,138,139,140]). Intriguingly, back in the 1970s, a number of psychiatric hospitals in North American would apparently play ‘Sea gulls…Music for rest and relaxation’ [141] for just this reason [142].

### 4.2. Hunger and Forgetting to Eat

Research conducted with individuals suffering from amnesia suggests that it is external cues that often trigger the initiation of meal consumption in the absence of awareness/memory of the meals that may just have been consumed [143]. At the same time, however, many of those older individuals living alone may simply forget to eat [144], because they lack the robust external (or exogenous) hunger cues, such as the kitchen aromas of food cooking that may play an important part in encouraging the rest of us that it is time to eat [145].

In order to try and address the latter problem, Prof. Spence was involved as a consultant in a project a few years ago designed to try and help older individuals, specifically early-stage Alzheimers/dementia patients who might otherwise need to be hospitalized due to undernutrition [146], to retain their independence living at home for a little longer. The idea behind the ‘Ode’, as it is called, was to release familiar meal time-specific and culturally/age group-appropriate ambient food scents into the home three times a day. The hope was that this might help those who might otherwise forget to eat, to eat. The results of a small study suggested the efficacy of this award-winning plug-in food scent alarm clock device.

The six food aromas developed for the launch included fresh orange juice, cherry Bakewell tart, homemade curry, pink grapefruit, beef casserole, and Black Forest gateau. They were specifically chosen to be representative of food aromas that were likely to be familiar to those in the target age group (though see also [147]). The results of a small-scale 10-week pilot study involving fifty people with dementia, along with their families, revealed that more than half of those who used the device ended up maintaining their weight, or else showing a slight increase, as compared to an expected decline in weight that is so often seen in this group (e.g., [148,149]).

## 5. Age-Related Decline in Multisensory Integration and Attention

Flavour is undoubtedly one of the most multisensory of our everyday experiences [150], potentially engaging each and every one of our senses. Hence, over and above any loss of sensitivity in the individual senses that either help to set our flavour expectations, or else contribute directly to our flavour experiences, one might also ask what role, if any, a more central loss of multisensory integration, or information-processing abilities [151,152], might have for food perception/behaviour amongst the elderly. While there has been little research directly targeting this question with respect to the chemical senses, one can find a multitude or answers in the case of the higher spatial senses [153].

For example, Laurienti, Burdette, Maldjian, and Wallace [154] have argued that multisensory integration may actually become more important as we age. They suggest that multisensory integration can, in some sense at least, help to make up for the loss of sensitivity and responsiveness of the individual senses as they start their inevitable decline (see also [155]). In their study, Laurienti and colleagues had groups of younger and older participants make speeded detection responses to a random sequence of auditory, visual, and audiovisual target stimuli. While unisensory response latencies were shown to slow with increasing age, multisensory RTs were similarly fast in both age groups. At the same time, however, other researchers have argued that the ability to integrate multisensory cues declines with increasing age (see also [156,157]). In particular, the increased risk of falling that has been documented amongst the elderly has been put down, at least in part, to a failure to appropriately integrate vestibular and visual cues [158].

The authors are not, however, aware of any research that has specifically addressed the question of whether there is any central impairment affecting the multisensory integration of the flavour senses (namely, retronasal olfaction, gustation, and possibly also trigeminal inputs) with advancing years. Here, it is worth stressing that it is not only the neural sites of multisensory integration that differ between the chemosensory and the spatial senses, but also the very nature of the rules governing that integration (see [159], for a review). So, for example, spatial co-location, and attention have been reported to play more of a modulatory role over the integration of auditory, visual, and tactile stimuli than would appear to be the case for the integration of the flavour senses.

Visual (colour) cues, in particular, exert a significant modulatory effect over the sensory-discriminative and hedonic aspects of tasting (see [160,161], for reviews). The visual dominance over taste/flavour perception as well as the central importance of visual cues for driving our food selection behaviours both need to be recognised (see [162]). Hence, one might consider whether the presentation of more brightly-coloured foods/dishes could be used to help stimulate the appetite amongst the elderly ([16]; though see also [163], on the changing patterns of sensory dominance with aging). Note here also how the presentation of a visually-stimulating array of produce colours also fits with contemporary nutritional guidelines [164,165].

Attention plays a key role in both the phenomenon of oral referral [166,167,168] and multisensory flavour perception more generally [169]. Furthermore, in younger people, the research clearly shows that increasing the perceptual load of a visual task lowers taste/flavour perception ([170]; see also [171]). Hence, one danger might be that the loss of flexibility of attentional allocation/switching, in the elderly [172] might mean that the TV dinner has an even more detrimental effect on the perception of food consumption-related sensory cues than on the rest of us [79].

## 6. Dining as a Fundamentally Social Activity

However, over and above any perceptual decline, a large part of the poor food consumption behaviours that have been evidenced in older populations are likely to result, at least in part, from the increasingly isolated living that many older people face, and the consequent lack of social interaction that have been documented amongst older age groups in many countries [79]. To give some sense of the problem in relation to food consumption, in Japan, where people live longer than in most other places, it has been estimated that 24% of pensioners eat the majority of their meals alone [173]. Importantly, here in the UK, eating meals alone has recently been reported to be the biggest lifestyle cause of unhappiness [174,175].

Recognizing this growing social issue, there is an emerging interest in the field of digital commensality, with a number of researchers trying to bring back the enjoyment of eating by means of digital technologies [176]. While acknowledging the lack of familiarity with contemporary digital technologies amongst many in this age group currently, there has nevertheless been widespread interest in the possibility of using robotic dining assistants to interact with elderly patients, simultaneously monitoring their consumption, and ideally nudging them toward healthier patterns of food consumption. Assisted eating at mealtimes in hospitals also helps here, though, as has already been mention, the cost implications cannot be ignored. This kind of approach recognizes the fact that both physiological and psychological factors play an important role in controlling eating in later life [177]. Of course, the social solutions in this space need not be digital, with other commentators advocating for lunch clubs [178]. Of perhaps more widespread relevance, given the rapid rise of at home food delivery services [79,179], there seems to be an unrecognized opportunity amongst the larger providers of food direct to the home (be it take-away or ready-made meals, or even meal kits, from the likes of Deliveroo, Uber Eats, or Blue Apron) to connect those individuals who may be living, (cooking) and eating alone over the internet with other like-minded individuals who find themselves in the same position [176]. There is also scope to enhance the provision of meals for one, too [180,181].

## 7. Conclusions

A wide body of evidence now points toward the conclusion that the rapidly-growing aging population are engaging in a variety of unhealthy eating behaviours. On the one hand, this takes the form of adding excessive amounts of sugar and salt to taste [66], while, at the same time, not eating enough to maintain a healthy weight. Indeed, clinical malnutrition is not uncommon amongst hospitalised elderly individuals, not to mention those living in care homes [71,72,75]. While sensory decline, specifically amongst the chemical senses, is undoubtedly partly to blame, the absence of social interaction is also likely to be a core part of the problem [176]. Indeed, one of the key challenges for the future is how to deal with the sensory underload so often experienced by those of advanced years [182], many of whom are living alone and eating alone far more often than they would like. Optimizing food design to stimulate the remaining functional senses (e.g., [73,96,183,184] and harnessing various digital technologies to increase the opportunity for distributed social interaction at mealtimes (and beyond) are all likely going to help manage the situation in the years ahead [176]. Looking to the future, it will likely become increasingly important to consider the changing role of the senses in food choice and food intake across the lifespan [87]. Furthermore, the idea of transgenerational design will likely become increasingly important too [149,185,186], especially given the increasing value of the silver dollar [187,188]. The importance of marketing foods effectively to this growing demographic should not be underestimated either [189].

As stressed in this review, one promising vehicle for the delivery of nutritional requirements in at least some elderly individuals may be nutritionally-enhanced ice-creams (see [110]; see also [190]). Such foods optimise food-related stimulation of the remaining functional senses, though the possibility of ‘disconfirmation of expectation’ given the novelty of such unusual formulations/flavours needs to be carefully thought through [130]. Although there is as yet limited research into this approach, the preliminary evidence at least looks promising [110]. Ultimately, it is only by optimizing the sensory, social, nutritional, and emotional aspects of food and eating design that we will be able to provide a satisfactory food environment in the years ahead.

Further, given the many reports of chemosensory loss constituting one of the most common symptoms of COVID-19 [191,192], a loss that has been reported to affect not only smell and taste, but also the trigeminal sense [193], one might wonder to what extent some of the sensory strategies outlined here to deal with the sensory losses experienced by the elderly might be relevant to providing enhanced food experiences for the worryingly large number of those suffering from long COVID. Indeed, according to the results of the latest report, approximately one-third of patients suffering from long COVID reported impaired chemosensory function three months after infection [194].

## Figures and Tables

**Figure 1 foods-10-00168-f001:**
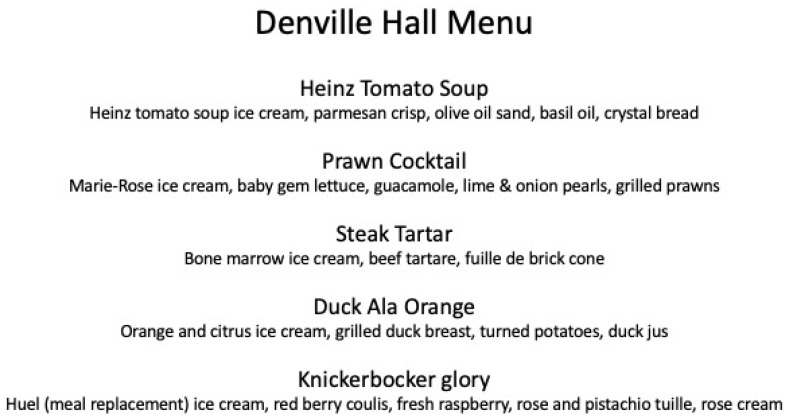
The menu created especially for the nostalgic flavours ice-cream-focused dining concept created for Denville Hall.

**Figure 2 foods-10-00168-f002:**
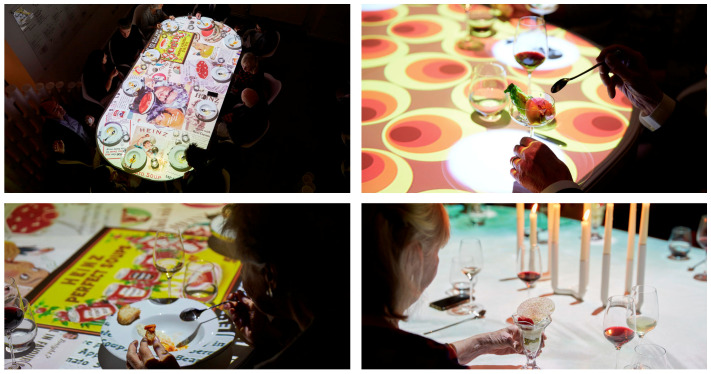
Still images from the Denville Hall dining concept.

## Data Availability

There is no data to share, though chef J.Y. woud be happy to share the recipes on request.

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
