# Peer review of "Aging and the (Chemical) Senses: Implications for Food Behaviour Amongst Elderly Consumers"

_foods, 2021, doi:10.3390/foods10010168_

Round 1

Reviewer 1 Report

This paper presents an analysis and overview of the decline of taste in aging populations. Authors examine multiple aspects: food composition (e.g. salt, spices, etc), smell (both orthonasal and retronasal), auditory cues, as well as the social aspect of eating alone or with others. The manuscript includes a very informative presentation of our current understanding of these factors, and the effect they have on the eating habits, malnutrition, and overall health of aging population.

The paper also presents a particular event where a five-dish course was served, however, all dishes were prepared as ice-cream flavored dishes based on the authors’ observation that ice-cream is a food type that is more easily accepted and consumed by the elderly, both because of its texture properties as well as of its taste.

My main and only concern is that there is no validation, or at least quantitative processing of the event’s data. While the participants have expressed positive views for the event, a more systematic presentation of the results would be beneficial. In general, the paper is well-written, easy to follow, and quite informative and interesting.

Author Response

Many thanks for the positive review.

The reviewer is right that there is no analysis of the dining event, and that is a limitation of the ice-cream intervention. Unfortunately, however, given the format of the event it was not possible to collect data, nor has it been possible to repeat the event subsequently.

Nevertheless, we believe that the suggested approach has value for the field.

In response to the reviewer, though, we have now made clearer in revision the anecdotal nature of the study mentioned, and stress the absence of quantitative evaluation.

Reviewer 2 Report

Overall, the paper is very well written and clear to be understood and follow. However, I have some small concerns.

In the title, it seems that will only be addressed the chemical senses, however, along with the manuscript information about visual (colour), and texture is provided. Therefore, I suggest the authors to broad the title.

I also suggest to update some of the references cited, as a lot of them are before the nineties, and nowadays, authors might find new evidence, especially in the first part of the manuscript. Even further, due to the current worldwide situation, why to not include the COVID19 loss of smell and taste? How to be aged and to have suffered COVID19 might affect? This might increase the attention and the number of reads and citations of the manuscript.

I do not know what has happened with the line numbers, but I will try to point out, some things that need to be changed using the page number.

  • Page 3, “Decline in salivary function with aging”- “Importantly, salivary function has been reported to declines…” Should the “s” be removed?
  • Page 3, “…this may also interfere with multisensory flavour perception in the elderly…” Not may, it does, as saliva not only helps in breaking down the food, increasing the flavour release, but i is the carrier of those chemical stimuli of what the authors are referring to.
  • Page 4, “The metallic taste that many patients report while undergoing treatment for cancer would also appear to be an effective appetite suppressant” But this, it is not only in the elderly, right? Is cancer more frequent in the elderly and that is why is cited in this manuscript?
  • Page 4, “One route to increasing the sensory/perceptual interest of food and drink amongst older populations is to enhance the stimulation provided by the residual senses”…what is the residual sense?
  • Page 9. “I am not, however, aware…” as the paper is written by two authors…should be.. “We are not…”?

Author Response

Many thanks for this positive review of our manuscript.

Overall, the paper is very well written and clear to be understood and follow. However, I have some small concerns.

MANY THANKS FOR THIS POSITIVE AND CONSTRUCTIVE REVIEW

In the title, it seems that will only be addressed the chemical senses, however, along with the manuscript information about visual (colour), and texture is provided. Therefore, I suggest the authors to broad the title.

GOOD POINT. TITLE REVISED IN REVISION.

I also suggest to update some of the references cited, as a lot of them are before the nineties, and nowadays, authors might find new evidence, especially in the first part of the manuscript. Even further, due to the current worldwide situation, why to not include the COVID19 loss of smell and taste? How to be aged and to have suffered COVID19 might affect? This might increase the attention and the number of reads and citations of the manuscript.

INDEED. In response, we have now updated the references to include a few more recent papers. We have also included a short section at the end concerning the loss of smell and taste in Covid, though note that this is not really a primary theme of our research. HENCE WHILE UNDOUBTEDLY A CITABLE ISSUE, IT IS NOT DIRECTLY RELEVANT TO THE QUESTION OF AGING THAT IS THE SUBJECT MATTER OF THIS SPECIAL ISSUE.

  • Page 3, “Decline in salivary function with aging”- “Importantly, salivary function has been reported to declines…” Should the “s” be removed?
  • INDEED CORRECTED THANKS
  • Page 3, “…this may also interfere with multisensory flavour perception in the elderly…” Not may, it does, as saliva not only helps in breaking down the food, increasing the flavour release, but i is the carrier of those chemical stimuli of what the authors are referring to.
  • MORE ASSERTIVE WORDING INCOPRORATED AS SUGGESTED
  • Page 4, “The metallic taste that many patients report while undergoing treatment for cancer would also appear to be an effective appetite suppressant” But this, it is not only in the elderly, right? Is cancer more frequent in the elderly and that is why is cited in this manuscript?
  • WHILE IT IS CERTAINLY TRUE THAT CANCER CAN STRIKE THOSE OF ANY AGE, THE INCIDENCE DOES INCREASE WITH AGE. A REFERENCE TO THAT EFFECT HAS NOW BEEN ADDED IN REVISION
  • Page 4, “One route to increasing the sensory/perceptual interest of food and drink amongst older populations is to enhance the stimulation provided by the residual senses”…what is the residual sense?
  • THE RESIDUAL SENSES REFERS TO THOSE NOT CONSTITUTIVELY INVOLVED IN TASTE/FLAVOUR PERCEPTION. WORDING MODIFIED IN REVISION TO MAKE CLEAR
  • Page 9. “I am not, however, aware…” as the paper is written by two authors…should be.. “We are not…”?
  • YES INDEED, CORRECTED.

THUS, ALL OF THESE HELPFUL SUGGESTIONS HAVE BEEN DEALT WITH IN REVISION.